# Lipoprotein(a): Evidence for Role as a Causal Risk Factor in Cardiovascular Disease and Emerging Therapies

**DOI:** 10.3390/jcm11206040

**Published:** 2022-10-13

**Authors:** Harpreet S. Bhatia, Michael J. Wilkinson

**Affiliations:** Division of Cardiovascular Medicine, Department of Medicine, University of California San Diego, La Jolla, CA 92093, USA

**Keywords:** lipoprotein(a), cardiovascular disease, risk factors, prevention

## Abstract

Lipoprotein(a) (Lp(a)) is an established risk factor for multiple cardiovascular diseases. Several lines of evidence including mechanistic, epidemiologic, and genetic studies support the role of Lp(a) as a causal risk factor for atherosclerotic cardiovascular disease (ASCVD) and aortic stenosis/calcific aortic valve disease (AS/CAVD). Limited therapies currently exist for the management of risk associated with elevated Lp(a), but several targeted therapies are currently in various stages of clinical development. In this review, we detail evidence supporting Lp(a) as a causal risk factor for ASCVD and AS/CAVD, and discuss approaches to managing Lp(a)-associated risk.

## 1. Introduction

Lipoprotein(a) (Lp(a)) is a lipid-carrying particle composed of a low-density lipoprotein (LDL)-like particle containing apolipoproteinB-100 (apoB) linked by a disulfide bond to apolipoprotein(a) (apo(a)). Apo(a) contains varying numbers of three-dimensional structures called kringles [1]. Lp(a) is primarily genetically determined through the *LPA* gene, and variants in the *LPA* gene are associated with cardiovascular disease [2,3]. Elevated Lp(a) is highly prevalent, occurring at levels >30 mg/dL in an estimated 35% of individuals and at levels >50 mg/dL in 24% of individuals [4]. Smaller isoforms of apo(a) are associated with higher Lp(a) levels. Importantly, the apo(a) isoform size and Lp(a) levels vary by ethnicity [5].

The normal physiological function of Lp(a) is unknown [1]. However, Lp(a) is associated with increased risk for several cardiovascular diseases (CVD), including coronary artery disease (CAD)/atherosclerotic cardiovascular disease (ASCVD) [2], aortic stenosis/calcific aortic valve disease (AS/CAVD) [6], ischemic stroke [7,8], heart failure [9], atrial fibrillation [10], and peripheral arterial disease [11]. Lp(a) is associated with risk for CAD through multiple mechanisms (Figure 1) including atherogenesis mediated by apoB [12], vascular inflammation mediated by its carriage of oxidized phospholipids (OxPL) [13,14,15,16], and anti-fibrinolytic effects that may be related to the homology of apo(a) with plasminogen [17]. Lp(a) is associated with risk for AS/CAVD through the pro-inflammatory and pro-calcification effects of OxPL that are likely able to enter the aortic valve through binding by apo(a). Lipoprotein-associated phospholipase A_2_ (Lp-PLA_2_) and autotaxin, enzymes present on Lp(a), are also likely involved in the pathogenesis of AS/CAVD [18]. This review will summarize the evidence for Lp(a) as a causal risk factor for ASCVD, as well as current and emerging therapies for elevated Lp(a).

International society guidelines differ in their recommendations for Lp(a) testing; however, multiple international societies recommend testing in all individuals. The 2019 American College of Cardiology (ACC)/American Heart Association (AHA) Primary Prevention Guideline [19] and the 2018 Multi-Society Cholesterol Guideline [20] both recommend using Lp(a) as a risk enhancer to guide therapy among borderline and intermediate risk individuals, particularly among individuals with a family history of premature CVD. The European Society of Cardiology (ESC)/European Atherosclerosis Society (EAS) dyslipidemia guideline recommends measurement of Lp(a) once in every adult’s lifetime [21]. Similarly, the Canadian Cardiovascular Society (CCS) dyslipidemia guideline recommends testing once for all individuals and use of Lp(a) as a risk modifier [22]. There is a need for standardization of Lp(a) measurement as there are multiple different assays and methods available [23]. Lp(a) measurement using an isoform-insensitive assay that is reported in nanomoles per liter (nmol/L) is recommended [24]. A recent study described a novel method for directly measuring Lp(a) cholesterol (Lp(a)-C) which also enables correction of LDL-cholesterol (LDL-C) as standard methods for measuring LDL-C also include Lp(a)-C [25].

## 2. Lipoprotein(a) and Risk for Atherosclerotic Cardiovascular Disease

Lp(a) is well established as a likely causal risk factor for ASCVD based on epidemiologic and genetic studies (Table 1).

### 2.1. Epidemiologic Studies of Lp(a) and Risk for ASCVD

Multiple prospective and epidemiologic studies including different populations have demonstrated an association between Lp(a) and various ASCVD outcomes. In a study of individuals with myocardial infarction (MI)/coronary heart disease (CHD) and controls from the Reykjavik Study, the top tertile of Lp(a) was independently associated with CHD risk (OR 1.60, 95% CI 1.38–1.85) [29]. In a large meta-analysis published in 2009, the Emerging Risk Factors Collaboration evaluated over 30 prospective studies including over 120,000 participants and demonstrated an association between higher Lp(a) levels and CHD and ischemic stroke [26]. A meta-analysis of 11 studies of individuals with CVD demonstrated an association between Lp(a) and CVD risk (highest quintile of Lp(a) OR 1.40, 95% CI 1.15–1.71) [27].

The association between Lp(a) and ASCVD risk has been shown across several different populations. In the Copenhagen City Heart Study (CCHS), Lp(a) was associated with MI in both men and women in a stepwise manner. In women, the 95th percentile of Lp(a) was associated with an HR of 3.6 (95% CI 1.7–7.7); in men, it was associated with an HR of 3.7 (95% CI 1.7–8.0) [30]. In Black and White participants from the Atherosclerosis Risk in Communities (ARIC) Study, Lp(a) levels were positively associated with CVD (coronary heart disease and ischemic stroke) events in a graded manner [31]. In a more recent study of seven ethnic groups, Lp(a) >50 mg/dL was associated with risk for myocardial infarction (overall OR 1.48, 95% CI 1.32–1.67). Again, a graded association between Lp(a) levels and outcomes was seen. Elevated Lp(a) was associated with increased risk in Chinese, European, Latin American, South Asian, and Southeast Asian individuals, but not in African or Arab individuals. The greatest population-attributable risk was noted in those of South Asian and Latin American descent [32]. In a very large recent study, however, the association between Lp(a) and CVD events was similar in White, South Asian and Black individuals, despite marked differences in median levels within these ethnic groups [34]. Lp(a) ≥ 50 mg/dL is also associated with increased risk for MI, stroke, and cardiovascular mortality in those with diabetes mellitus as well as pre-diabetes, with a graded association noted from Lp(a) 30–50 mg/dL and Lp(a) ≥50 mg/dL [33]. In a meta-analysis of statin outcome trials using individual patient level data from over 29,000 participants, Lp(a) was associated linearly with risk for CVD at baseline and on statin therapy [28]. Key features of these epidemiologic studies are the graded association between Lp(a) and events, suggestive of a true biological phenomenon, and the relative consistency across groups, including diverse racial/ethnic groups.

### 2.2. Genetic Studies of Lp(a) and Risk for ASCVD

Genetic studies have been critical in establishing Lp(a) as a likely causal risk factor for ASCVD with a robust evidence base. In a large genetic study of over 48,000 single-nucleotide polymorphisms (SNPs) from 2100 genes in over 3000 participants with coronary disease and over 3000 controls, the region of the *LPA* gene had the strongest association with coronary disease. In particular, the rs10455872 and rs3798220 SNPs were identified and both were associated with increased Lp(a) levels and with positive odds ratios for CAD of 1.70 (95% CI 1.49–1.95) and 1.92 (95% CI 1.48–2.49), respectively [2]. In another study, both plasma Lp(a) levels and Lp(a) kringle IV type 2 (KIV-2) size polymorphism genotype were associated with risk for MI [35]. In a genome wide association study (GWAS) of CAD in >60,000 CAD cases and >130,000 controls, the *LPA* SNP rs3798220 was again associated with CAD with an OR of 1.28 (*p* < 0.001) [3]. A prospective study of >8000 Danish individuals demonstrated that the addition of Lp(a) levels ≥80^th^ percentile, number of KIV-2 repeats, and carrier status for the *LPA* SNP rs10455872 improved MI and coronary heart disease (CHD) risk prediction in addition to traditional risk factors [36]. In a case-control study of individuals with CAD, an *LPA* null allele (rs41272114) was evaluated. The null allele was associated with decreased Lp(a) levels, as well as decreased CAD risk, compared to noncarriers [37]. Another study demonstrated that splice variants in *LPA*, associated with reduced Lp(a) levels, were protective against cardiovascular disease [38]. *LPA* SNPs have also been shown to vary by ethnicity [5]. A recent Mendelian randomization study observed an OR per 1-SD increment in Lp(a) of 1.10 (95% CI 1.05–1.14) for MI [39].

The risk associated with Lp(a) has also been evaluated recently in the context of other risk factors. Lp(a) is independently associated with CVD even when accounting for family history of CHD [40]. The use of apolipoproteinB100 (apoB) as a risk marker has been a source of considerable interest recently. In one study, the risk associated with Lp(a) persisted when adjusting for apoB, while the risk associated with LDL-C was attenuated. These results suggest that apoB does not sufficiently encompass Lp(a)-associated risk [41]. Finally, one study evaluated CVD risk associated with Lp(a) stratified by high-sensitivity C-reactive protein (hsCRP), given the inflammatory risk associated with Lp(a). In this study, the independent risk associated with Lp(a) was only present with elevated hsCRP levels; however, this requires further study [42].

Lp(a) has been identified as a risk factor for ASCVD in many epidemiologic studies, often in a dose-dependent fashion, suggesting a pathophysiologic mechanism. Genetic studies have strengthened the evidence for Lp(a) as a causal risk factor, particularly Mendelian randomization studies that reduce confounding.

## 3. Lipoprotein(a) and Risk for Aortic Stenosis/Calcific Aortic Valve Disease

The other disease with which Lp(a) is most often associated is aortic stenosis (AS), or calcific aortic valve disease (CAVD). A number of epidemiologic and genetic studies support the association between Lp(a) and aortic valve calcification, AS, and progression of AS (Table 2).

### 3.1. Epidemiologic, Imaging, and Mechanistic Studies of Lp(a) and Calcific Aortic Valve Disease

Lp(a) has been associated with aortic valve sclerosis for many years, raising suspicion for Lp(a) as a cause of AS. In 1995, the prevalence of aortic valve sclerosis was observed to increase in association with Lp(a) levels [43]. In the Cardiovascular Health Study (CHS), Lp(a) was also associated with increased risk for aortic valve sclerosis or stenosis [44]. Importantly, Lp(a) and Lp(a)-associated molecules (e.g., OxPL) have been detected in the AV leaflets of individuals with calcific AS [45].

Lp(a) has also consistently been associated with aortic valve calcification (AVC) through imaging and basic studies, which may link Lp(a) and AS pathophysiologically. In an echocardiographic study, Lp(a) levels were independently associated with AVC [46]. In asymptomatic individuals with familial hypercholesterolemia, Lp(a) was significantly associated with AVC by computed tomography (CT) [47]. In another study utilizing 18F-sodium fluoride positron emission tomography (PET)/CT, elevated Lp(a) was associated with AV microcalcification even before the development of clinical AS [49]. Another PET/CT study similarly demonstrated that higher Lp(a) and OxPL levels were associated with increased AV calcification activity [50]. Autotaxin, transported by Lp(a) to the aortic valve, also promotes inflammation and calcification of the aortic valve [48].

In 2003, a study of individuals with severe AS and age-matched controls observed an association between elevated Lp(a) (≥48 mg/dL) and risk for AS [51]. In a very large study of two prospective cohort studies, Lp(a) was significantly associated with AS in a dose-dependent fashion [52]. In the European Prospective Investigation into Cancer (EPIC)-Norfolk Study, the top tertile of Lp(a) levels was associated with increased risk for AS [53]. In another large study, each standard deviation increase in Lp(a) was associated with an HR of 1.23 (95% CI 1.06–1.41) for AS [54].

Oxidized phospholipids, which are carried by Lp(a), are also implicated in AS, likely due to their role in inflammation and calcification [57]. In addition to Lp(a), OxPL are detected in the AV leaflets of individuals with calcific AS [45]. In a mouse model, inactivation of OxPL resulted in decreased AV calcification and reduced the development of AV gradients [56]. In humans, Lp(a) induces osteogenic differentiation of vascular cells, which is mediated by OxPL and inhibited by inactivating OxPL, again providing a possible mechanism for the link between Lp(a) and AS [50]. OxPL-apoB levels are also associated with risk for calcific aortic valve disease in a dose-dependent manner [55].

Lp(a) is also associated with faster progression of AS, which may be particularly meaningful clinically. In a study of individuals with mild-to-moderate AS in the ASTRONOMER trial, individuals in the top tertile of Lp(a) levels and OxPL-apoB levels had greater risk for rapid progression [57], and Lp(a) and OxPL-apoB levels were linearly associated with faster progression [58]. Higher Lp(a) and OxPL levels are also associated with increased progression of aortic valvular calcium score by CT and faster hemodynamic progression by echocardiography, as well as greater risk for aortic valve replacement and death [50].

### 3.2. Genetic Studies of Lp(a) and Calcific Aortic Valve Disease

Genetic studies have again been critical for establishing Lp(a) as a likely causal risk factor for AS and the only monogenic risk factor for AS. In a GWAS for AV calcification by CT, the *LPA* SNP rs10455872 was identified as the only SNP to meet genomewide significance. The *LPA* genotype was also associated with incident AS and the need for AV replacement [6]. In a Mendelian randomization study, the rs10455872 variant was also strongly associated with increased risk for AS with greater risk among homozygous carriers than heterozygous [53]. In a study incorporating multiple SNPs, genotypes corresponding with Lp(a) levels were associated with an increased risk for AS [52]. In another Mendelian randomization study, the *LPA* SNPs rs3798220 and rs10455872 and the *LPA* KIV-2 genotype were associated with AS [54].

In conclusion, a large body of evidence has established Lp(a) as a likely causal risk factor for calcific aortic valve disease and AS. Lp(a) and OxPL are associated with aortic valve calcification, even before the development of clinical AS, and are found in calcified aortic valve leaflets. *LPA* gene variants are similarly associated with calcification. Clinically, Lp(a), OxPL, and *LPA* variants are associated with the incidence of AS, in a dose-dependent manner, as well as the risk for progression of AS.

## 4. Current Therapies and Lipoprotein(a)

While there are no medications specifically approved in the United States for risk associated with elevated Lp(a), a number of currently available therapies have been evaluated (Table 3). Statins are a cornerstone of therapy for prevention of cardiovascular disease. However, statin therapy does not lower Lp(a) and may even increase it [59]. Of particular importance is that CVD risk associated with elevated Lp(a) persists in statin-treated patients with an HR of 1.43 (95% CI 1.15–1.76) for Lp(a) ≥ 50 mg/dL in statin-treated patients compared with an HR of 1.31 (1.08–1.58) prior to statin initiation in statin clinical trials [28]. Additionally, statins have not shown a benefit for reducing the progression of AS [60], and this may be partially explained by their lack of effect on Lp(a). Thus, while statins are an important therapy for primary and secondary prevention of CVD, they do not address Lp(a) levels and Lp(a)-mediated risk. Ezetimibe has also been evaluated, resulting in a 3% decrease in Lp(a) at 12 weeks [61] in one study and a 29% reduction at 12 weeks in another [62]. However, these data are limited, particularly in regard to their impact on outcomes.

Niacin has been shown to lower Lp(a) levels but without a clear impact on cardiovascular outcomes. In the AIM-HIGH (Atherothrombosis Intervention in Metabolic Syndrome with Low HDL/High Triglyceride and Impact on Global Health Outcomes) trial, participants with low baseline HDL-C and CVD were randomized to simvastatin plus placebo or simvastatin plus niacin with ezetimibe if needed. Niacin reduced Lp(a) by 21% at 1 year, but Lp(a)-associated risk remained with an on-study HR of 1.18 (*p* = 0.03) compared with a baseline HR of 1.25 (*p* = 0.001) [64]. In the HPS2-THRIVE study (Heart Protection Study 2-Treatment of HDL to Reduce Incidence of Vascular Events), individuals with vascular disease were randomized for extended release niacin and laropiprant (to reduce the side effects of niacin) or placebo. There was no overall effect on vascular disease, but there was a modest absolute reduction in Lp(a) [65]. It should be noted that in both studies, the absolute reduction in Lp(a) was low, and the trials were not designed to assess the impact of niacin on CVD risk in elevated Lp(a). Additionally, these trials highlight the potential risks for significant side effects from niacin.

Lipoprotein apheresis, through multiple available techniques, is very effective at lowering Lp(a) levels, with an acute reduction up to 75% and a reduction in mean concentrations between sessions of up to 40% [71]. A retrospective study in the U.S. of 14 patients with CVD and elevated Lp(a) (mean 138 mg/dL) with normal LDL-C (mean 80 mg/dL) observed a reduction of 63% in Lp(a) and 64% in LDL-C with lipoprotein apheresis, translating into a 95% reduction in MACE over 48 months [72]. In Germany, a prospective study of 170 patients with CVD and mean LDL-C 99 mg/dL and Lp(a) of 108 mg/dL observed a reduction in Lp(a) of 68% with a single treatment, and a reduction in the MACE annual event rate from 0.58 to 0.11 with regular lipoprotein apheresis [73]. These studies demonstrate that lipoprotein apheresis is very effective in reducing Lp(a) levels, which may translate into a reduction in CVD events, but data are limited. Lipoprotein apheresis is currently the only FDA-approved therapy for elevated Lp(a), but further study is needed. MultiSELECt (A European Multicenter Study on the Effect of Lipoprotein(a) Elimination by Lipoprotein Apheresis on Cardiovascular outcomes) is an ongoing prospective cohort study to evaluate the effect of lipoprotein apheresis on events in individuals with elevated Lp(a) [74].

Proprotein convertase subtilisin/kexin type 9 inhibitors (PCSK9i) are one of the most promising therapies currently available for addressing Lp(a)-associated risk. In an analysis from the FOURIER (Further Cardiovascular Outcomes Research with PCSK9 Inhibition in Subjects with Elevated Risk) trial, evolocumab reduced Lp(a) levels by a median of 27%. In those with Lp(a) above the median, there was a more potent reduction in events with an absolute risk reduction of 2.5% compared to 1.0%. There was an estimated 15% lower risk per 25 nmol/L reduction in Lp(a) with adjustment for the change in LDL-C [67]. In an analysis of the ODYSSEY Outcomes (Evaluation of Cardiovascular Outcomes After an Acute Coronary Syndrome During Treatment with Alirocumab) trial, alirocumab reduced Lp(a) by 23%, and a 1 mg/dL reduction in Lp(a) was associated with an HR of 0.994 (*p* = 0.008) [68]. Additionally, baseline Lp(a) levels predicted the risk reduction with alirocumab with a greater reduction in risk with increasing quartile of Lp(a) [88]. PCSK9i also significantly lower Lp(a) levels in addition to background niacin therapy [89]. Taken together, these studies suggest that PCSK9i reduce Lp(a) levels modestly, and the reduction in Lp(a) potentially translates into risk reduction independent of LDL-C reduction. A newer PCSK9i siRNA, inclisiran, was also shown to reduce Lp(a) by 19–22% in the ORION-10 and ORION-11 trials, but these trials were not designed to evaluate the effects of inclisiran on MACE [69]. Despite modest Lp(a) lowering, however, individuals treated with PCSK9i have evidence of residual vascular inflammation [90]. A recent study demonstrated that PCSK9i did not lower OxPL, despite Lp(a)-lowering, which may partially explain this residual inflammatory risk [70]. Recent society guidelines have incorporated the use of PCSK9i into recommendations for management of individuals with elevated Lp(a). The European Society of Cardiology (ESC)/European Atherosclerosis Society (EAS) dyslipidemia guidelines recommend consideration of PCSK9i in individuals with familial hypercholesterolemia and high Lp(a) (class IIa) [21], while the Canadian Cardiovascular Society (CCS) guidelines recommend consideration of PCSK9i for secondary prevention in individuals with Lp(a) ≥60 mg/dL [22].

Multiple other therapies have been evaluated with regard to Lp(a), again with unclear impact on outcomes. Mipomersen, an anti-sense oligonucleotide (ASO) apoB synthesis inhibitor, resulted in a reduction in Lp(a) by a median of 26% in four phase 3 trials of individuals with various hypercholesterolemic conditions [66]. However, in transgenic mice, free apo(a) levels were unaffected with mipomersen [91], and the impact of these findings on outcomes is unclear. Lomitapide, an inhibitor of microsomal triglyceride transfer protein (MTP) that transfers lipids to apoB, resulted in a 16% reduction in Lp(a) in one trial, but the impact on clinical outcomes was not evaluated [61]. Mipomersen and lomitapide are approved for individuals with homozygous familial hypercholesterolemia [92], but their use is limited by potential liver toxicity.

Finally, non-lipid therapies have also been studied for addressing Lp(a)-associated risk. Anti-platelet therapies have been evaluated for both primary and secondary prevention given Lp(a)’s association with coagulation and platelet aggregation pathways. For primary prevention, aspirin was studied in a secondary analysis of the Women’s Health Study, which randomized healthy women to aspirin 100 mg every other day vs. a placebo. In over 25,000 White women who were genotyped, aspirin was associated with a dramatic reduction in CVD events among carriers of the *LPA* rs3798220 SNP (which was associated with 2-fold increased CVD risk in the placebo group) with an HR of 0.44 (95% CI 0.20–0.94). Aspirin use was not associated with a reduction in risk among non-carriers [75]. However, this SNP was only present in 3.7% of individuals, and the results are only generalizable to White women. A recent analysis of White participants in the ASPREE (Aspirin in Reducing Events in the Elderly) trial of aspirin for the primary prevention of CVD in healthy elderly individuals demonstrated similar findings. Carriers of the rs3798220-C variant or individuals with high genetic risk based on a genetic risk score had significantly increased risk of MACE in the placebo group, but not in the aspirin group, again suggesting that aspirin may benefit individuals with increased genetic risk associated with Lp(a) [76]. Further study is needed to evaluate the use of aspirin in association with plasma Lp(a) levels and in a broader population with modern background therapy. In terms of secondary prevention, a study of patients with CAD after PCI demonstrated that prolonged dual anti-platelet therapy (DAPT) >1 year resulted in a significant reduction in CVD events (HR 0.54, 95% CI 0.31–0.92) compared with DAPT ≤1 year [77], again suggesting that there is a role for specific considerations related to anti-platelet therapy in this population. Hormone replacement therapy (HRT) has been a source of historical interest as it has previously been shown to lower Lp(a) levels. However, in a recent study of post-menopausal women, HRT resulted in a small reduction in Lp(a) levels, but did not result in a reduction in CHD events [79]. L-carnitine has also been associated with a reduction in Lp(a) levels. In a meta-analysis of randomized controlled trials, L-carnitine was associated with a mean reduction in Lp(a) levels of 8.8 mg/dL (95% CI −10.1, −7.6 mg/dL), particularly with the oral formulation. However, the impact of this modest reduction on clinical events was not evaluated [80]. In addition, L-carnitine is associated with accelerated atherosclerosis and increased CVD risk [81].

Lifestyle interventions, while integral to general cardiovascular health, have not been shown to have a significant impact on Lp(a). A thorough review of non-genetic influences on Lp(a) levels was recently published [93]. In one study, intensive multifactorial lifestyle intervention including diet, exercise, and smoking cessation did not result in a change in Lp(a) levels. [94]. Several studies have observed changes in Lp(a) levels with various diets; however, the changes are almost universally modest. Diet has been shown to modestly influence Lp(a) levels. In general, Lp(a) levels do not vary significantly whether in a fasting or nonfasting state [95]. The composition of macronutrients in diet does appear to influence Lp(a) levels. A low carbohydrate diet resulted in a nearly 15% reduction in Lp(a) in one study [96]. Low and moderate fat diets have also been shown to result in modest reductions in Lp(a) [97]. Carbohydrate intake may have a greater influence on Lp(a) levels than fat intake, as a low-fat, high-carbohydrate diet increased Lp(a) levels compared to a high-fat, low-carbohydrate diet [98]. High-carbohydrate and high-protein diets increase Lp(a) more than high unsaturated-fat diets, but all the absolute increases are small [99]. The type of dietary fat also appears to be relevant. Reducing dietary saturated fat results in a small increase in Lp(a) levels [100], and replacement of saturated fat with monounsaturated fat intake results in an increase in Lp(a) levels (11%), but less so than replacement with carbohydrates (20%) [101]. The data are not entirely consistent, as a Mediterranean-style diet with increased monounsaturated fatty acids decreased trans-fat, increased protein, and decreased carbohydrate intake from baseline resulted in a significant decrease in Lp(a) levels; however, mean levels were not elevated to begin with [102]. In addition, alcohol consumption does not appear to be associated with Lp(a) levels [103]. Finally, physical activity does not appear to have a significant impact on Lp(a) levels [93].

## 5. Emerging Therapies to Lower Lipoprotein(a)

Targeted therapies to lower lipoprotein(a) are currently under development, with safety and efficacy being testing across phase I-III clinical trials (Table 3). Apo(a), as a key component of Lp(a), is the target of RNA-based therapeutics currently in phase II-III trials. The inhibition of apo(a) synthesis at the RNA level is a highly effective means of potently lowering circulating levels of Lp(a). Two methods of inhibiting the apo(a) mRNA have been Id: anti-sense oligonucleotides (ASO) and small-interfering RNA (siRNA). While different in their mechanism of targeting the apo(a) mRNA and promoting its degradation, both ASO and siRNA molecules are active in hepatocytes where their activity inhibits production of the downstream apo(a) protein and thus assembly of Lp(a) particles. A comprehensive review of the development and mechanism of these RNA-based therapeutics for Lp(a) was recently published elsewhere [92]. Here, we will summarize the key findings with regard to safety and efficacy of these emerging therapies.

Pelacarsen is an ASO targeting apo(a) administered by subcutaneous injection. The molecule has changed over time, for example, with the addition of ligand conjugation with N-acetylgalactosamine (GalNac), improving hepatocyte uptake, and allowing for lower doses of the drug to be administered. In a phase I study of a shorter-acting (second-generation) apo(a) ASO, 47 healthy volunteers with Lp(a) of at least 25 nmol/L were treated with either single dose or multiple (six) doses of the study drug vs. placebo. A single dose did not significantly lower Lp(a) at day 30, but a dose-response effect was observed after multiple doses at 36 days, up to a 77.8% reduction in Lp(a) from baseline (*p* = 0.001). Mild injection site reactions occurred [82]. This second-generation ASO was subsequently studied in a phase II trial of participants with elevated Lp(a), while the next generation ASO (ligand conjugated) entered phase I. The ligand-conjugated form demonstrated significant dose-dependent reductions in Lp(a), at day 30 up to 92% (*p* = 0.0007 vs. placebo) [83]. The safety and efficacy of this ligand-conjugated form of the apo(a) ASO led to its further development and testing in a phase II trial. In phase II, this hepatocyte-directed form of the apo(a) ASO was studied in participants with a history of ASCVD and baseline Lp(a) of at least 150 nmol/L. At 6 months, a dose-dependent reduction in Lp(a) was observed with up to a 80% reduction for 20 mg administered weekly. Injection site reactions were the most common adverse events [84]. The equivalent to this 20 mg/week formulation was chosen for the phase III cardiovascular outcomes trial with pelacarsen, Lp(a) HORIZON, which is currently underway and studying the impact of 80 mg/month of pelacarsen vs. placebo on rates of recurrent ASCVD events in a secondary prevention population with baseline elevated Lp(a) (NCT04023552).

Olpasiran is a GalNac-conjugated siRNA targeting apo(a) administered by subcutaneous injection. In a phase I, single-ascending-dose study of olpasiran vs. placebo in participants with Lp(a) either ≥ 70 and ≤199 nmol/L (n = 40) or ≥200 nmol/L (n = 24), the maximum mean percent change in Lp(a) from baseline ranged from −71% to −97%. Of note, the maximum reduction in Lp(a) was observed between days 43 and 71. While Lp(a) levels gradually increased, they remained lower compared to the placebo group out to 225 days. Olpasiran was well-tolerated [85]. A phase II clinical trial with olpasiran (OCEAN(a)-DOSE) is currently underway involving participants with Lp(a) > 150 nmol/L and history of ASCVD (NCT04270760). Additional siRNA therapies targeting apo(a) mRNA are under development. In a phase I clinical trial of SLN360 (an siRNA targeting apo(a) mRNA) vs. placebo, participants with Lp(a) ≥ 150 nmol/L were treated with single-ascending doses administered by subcutaneous injection. The maximal median percent reduction in Lp(a) was dose-dependent, up to −98%. The drug was generally well-tolerated; however, injection site reactions were reported [86].

Other targeted therapies for Lp(a) are in the early stages of development, including another siRNA, LY3819469, administered subcutaneously (NCT04914546). A phase I study of LY3473329, an oral medication targeting Lp(a), is also underway (NCT04472676). Thus, there is great interest in the continued development of compounds that can safely and potently lower Lp(a).

## 6. Conclusions and Future Directions

Lp(a) is now well-established as a risk factor for ASCVD and calcific aortic valve disease. However, optimal management of individuals with elevated Lp(a) is not well-established. Several currently available therapies have been evaluated for use in individuals with elevated Lp(a). However, improvement in clinical outcomes has only been shown in post hoc analyses from PCSK9i cardiovascular outcomes trials and in uncontrolled studies involving lipoprotein apheresis. There may also be an expanded role for anti-platelet therapy in both primary and secondary prevention in individuals with elevated Lp(a), but more research is needed. Multiple promising therapies that produce potent Lp(a) lowering are currently under investigation. Representative patient case scenarios are presented (Figure 2) to summarize an approach to management based on currently available evidence and guidelines. There are several areas in which future research is needed.

For secondary prevention of CVD, the biggest area of controversy is whether Lp(a)-lowering translates to reduced risk of ASCVD events, and what degree of Lp(a)-lowering is necessary to achieve this effect. Lp(a)HORIZON (NCT04023552) is currently underway and will evaluate the effect of potent Lp(a)-lowering with pelacarsen in the setting of secondary prevention. Another important question is whether prolonged dual anti-platelet therapy after revascularization improves outcomes in individuals with high Lp(a).

For primary prevention, a number of open questions remain. If Lp(a)HORIZON produces positive results, the natural extension may be a large, primary prevention trial to again evaluate if Lp(a)-lowering, and to what degree, will prevent CVD in primary prevention. The use of aspirin for primary prevention, again suggested to have benefit in the Women’s Health Study and ASPREE analyses, needs further evaluation. Another open area of investigation is the performance of current risk stratification tools in the context of elevated Lp(a), and whether Lp(a) should be incorporated into these tools. 

Another potential area for investigation will be aortic stenosis and whether Lp(a) lowering will prevent or halt the progression of aortic stenosis/CAVD. Finally, Lp(a) may partially explain residual inflammatory ASCVD risk. Further studies may evaluate whether targeting Lp(a)/OxPL reduces this inflammatory risk.

## Figures and Tables

**Figure 1 jcm-11-06040-f001:**
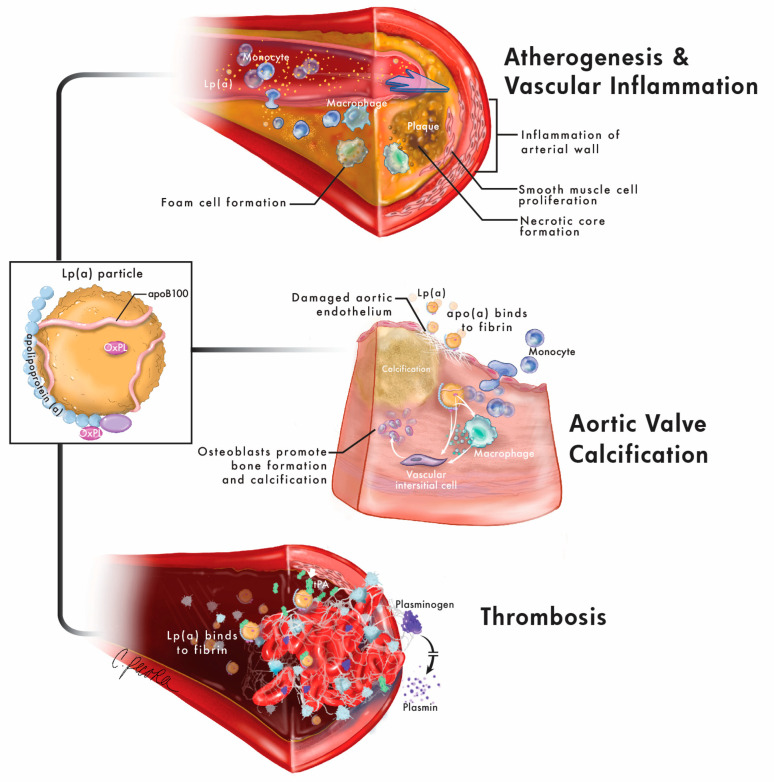
Mechanisms of cardiovascular risk related to Lp(a). Lipoprotein(a) and its individual components are associated with cardiovascular disease through multiple overlapping mechanisms. Lp(a) is composed of apolipoproteinB100 (apoB100) and apolipoprotein(a) (apo(a)), both of which contain oxidized phospholipids (OxPL). The apoB100 contributes to atherogenesis through similar mechanisms as low-density lipoprotein (LDL), including vessel wall binding, smooth muscle cell proliferation, foam cell formation and necrotic core formation. OxPL contribute to vascular inflammation through increased transmigration and cytokine production by monocytes as well as upregulation of inflammatory genes. Lp(a) contributes to aortic valve calcification as apo(a) binds to fibrin on injured aortic endothelium, and OxPL promote calcification and bone formation via vascular interstitial cells and upregulation of reactive oxygen species and proinflammatory cytokines in macrophages. Finally, apo(a) contributes to thrombosis by inhibiting fibrinolysis through competitive inhibition of tissue plasminogen activator (tPA) activation of plasminogen to plasmin and plasminogen binding to fibrin as well as promoting increased platelet activity.

**Figure 2 jcm-11-06040-f002:**
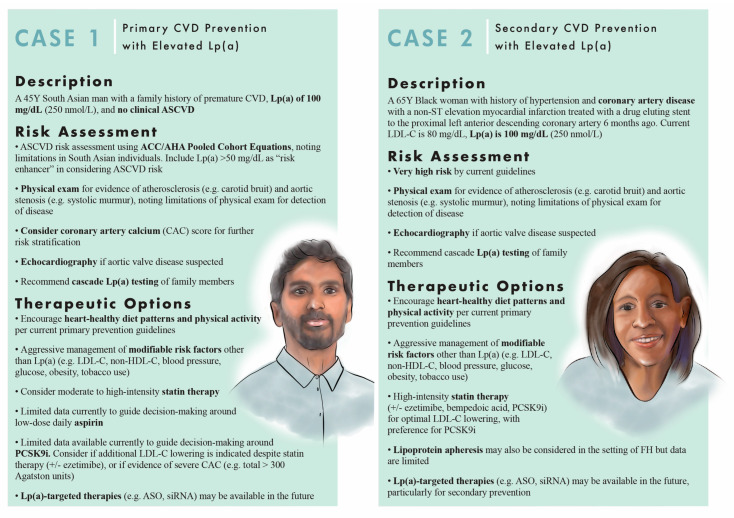
Patient cases: risk assessment and therapeutic options in the setting of elevated Lp(a). Two patient case scenarios are presented above, one for the primary prevention and one for secondary prevention in the setting of elevated Lp(a), with discussion of risk assessment and therapeutic options. **For case 1 (primary prevention),** the patient evaluation starts with risk assessment using the ACC/AHA Pooled Cohort Equations [19]. Lp(a) ≥50 mg/dL is considered a risk enhancer and would be an indication for more aggressive management of risk factors in an individual at borderline or intermediate 10-year risk. A physical exam is important to evaluate for evidence of atherosclerosis or aortic stenosis. If aortic valve disease is suspected, echocardiography would be indicated. CAC scoring can be considered to further risk stratify [20,22]. Cascade Lp(a) testing may be recommended for family members [104,105]. With regards to therapy, a healthy lifestyle should be recommended to all patients [19,106]. In the setting of elevated Lp(a), other modifiable risk factors should be addressed, and moderate to high-intensity statin therapy should be considered. Low dose daily aspirin may be a consideration, but there is currently limited data to guide this decision. Similarly, there is limited date for PCSK9i, but they may considered, particularly if additional LDL-C lowering is needed despite statin therapy, or there is evidence of severe CAC. Lp(a)-targeted therapies may be available as an option in the future. **For case 2 (secondary prevention),** the patient is considered very high risk by current guidelines [20]. Again, physical exam is important, and echocardiography is indicated if there is suspicion for aortic valve disease. Cascade Lp(a) testing may also be recommended for family members. Regarding therapy, a healthy lifestyle and management of other risk factors are again recommended. High intensity statin therapy should be prescribed, with consideration of PCSK9i if further LDL-C lowering needed [20,22]. Lipoprotein apheresis may be considered in the setting of FH [107]. Lp(a)-targeted therapies may also be an option in the near future, particularly for secondary prevention.

**Table 1 jcm-11-06040-t001:** Key studies of Lp(a) as a risk factor for ASCVD.

Meta-Analyses		
Author	Year	Key Findings
The Emerging Risk Factors Collaboration [26]	2009	Lp(a) associated with CHD (RR per SD 1.13, 95% CI 1.09–1.18) Ischemic stroke (RR per SD 1.10, 95% CI 1.02–1.18)
O’donoghue, et al. [27]	2014	Lp(a) associated with MACE in population with CAD: Highest quintile: OR 1.40, 95% CI 1.15–1.71
Willeit, et al. [28]	2018	Lp(a) associated linearly with CVD at baseline and on-statin therapy in statin outcomes trials On statins: Lp(a) 15–30 mg/dL: HR 0.95 (95% CI 0.82–1.11) Lp(a) 30–50 mg/dL: HR 1.08 (95% CI 0.95–1.23) Lp(a) ≥50 mg/dL: HR 1.42 (95% CI 1.16–1.74)
**Epidemiologic Studies**		
**Author**	**Year**	**Key Findings**
Bennet, et al. [29]	2008	Top tertile of Lp(a) associated with CHD with OR 1.60 (95% CI 1.38–1.85)
Kamstrup, et al. [30]	2008	Lp(a) associated with risk for MI in men and women: Women > 95th percentile: HR 3.6 (95% CI 1.7–7.7) Men > 95th percentile: HR 3.7 (1.7–8.0)
Virani, et al. [31]	2012	Highest quintile of Lp(a) associated with incident CVD risk in: Black individuals (HR 1.35, 95% CI 1.06–1.74) White individuals (HR 1.27, 95% CI 1.10–1.47)
Paré, et al. [32]	2019	Lp(a) > 50 mg/dL associated with increased risk of MI (OR 1.48, 95% CI 1.32–1.67) overall and in all ethnic groups studied except African and Arab individuals
Jin, et al. [33]	2019	Lp(a) ≥ 50 mg/dL associated with increased risk of CVD in: Pre-diabetes (HR 2.67, 95% CI 1.38–5.42) Diabetes (HR 3.47, 95% CI 1.80–6.69)
Patel, et al. [34]	2021	Lp(a) associated with increased ASCVD risk with HR 1.11 (95% CI 1.10–1.12) per 50 nmol/L increment Consistent results seen in White, South Asian, and Black individuals
**Genetic Studies**		
**Author**	**Year**	**Key Findings**
Clarke, et al. [2]	2009	*LPA* locus had strongest association with coronary disease in large study of candidate SNPs *LPA* SNP rs10455872 OR 1.70 (95% CI 1.49–1.95) for coronary disease *LPA* SNP rs3798220 OR 1.92 (95% CI 1.48–2.49) for coronary disease
Kamstrup, et al. [35]	2009	Higher levels of Lp(a) and lower number of kringle IV repeats associated with greater MI risk: >95th percentile of Lp(a): HR 2.6 (95% CI 1.6–4.1) 1st quartile of KIV-2 repeats: HR 1.5 (95% CI 1.2–1.9)
CARDIoGRAMplusC4D Consortium [3]	2013	*LPA* SNP rs3798220 associated with CAD: OR 1.28 (*p* < 0.001)
Kamstrup, et al. [36]	2013	Addition of Lp(a) levels, KIV-2 repeats and carrier status for *LPA* SNP rs10455872 to traditional risk factors all improved risk prediction for MI and CHD
Kyriakou, et al. [37]	2014	A null allele (*LPA* SNP rs41272114) was associated with decreased Lp(a) levels and decreased CAD risk
Lim, et al. [38]	2014	Splice variants of Lp(a) associated with reduced Lp(a) levels and protection against CVD (OR 0.84, *p* < 0.001)
Lee, et al. [5]	2016	Lp(a) levels and SNPs vary by ethnicity. The addition of SNPs to Lp(a) levels did not appear to be clinically meaningful.
Salaheen, et al. [39]	2017	OR per 1-SD increment of Lp(a) for MI 1.10 (95% CI 1.05–1.14)

CAD = coronary artery disease, CHD = coronary heart disease, CVD = cardiovascular disease, KIV = kringle 4, MACE = major adverse cardiovascular events, MI = myocardial infarction, SNP = single nucleotide polymorphism.

**Table 2 jcm-11-06040-t002:** Key studies of Lp(a) as a risk factor for calcific aortic valve disease.

Author	Year	Key Findings
Lp(a) and AV Sclerosis		
Gotoh, et al. [43]	1995	Greater prevalence of aortic valve sclerosis in individuals with Lp(a) ≥ 30 mg/dL (36.1%) compared with <30 mg/dL (12.7%, *p* < 0.001)
Stewart, et al. [44]	1997	Lp(a) associated with increased risk for aortic valve stenosis or sclerosis (OR 1.23, 95% CI 1.14, 1.32)
Torzewski, et al. [45]	2017	Lp(a) and associated molecules including OxPL detected in AV leaflets of individuals with calcific AS
**Lp(a) and AV Calcification**		
Bozbas, et al. [46]	2007	Lp(a) independently associated with AVC (OR 1.01, 95% CI 1.00–1.03)
Vongpromek, et al. [47]	2015	OR per 10 mg/dL increase in Lp(a) 1.11 (95% CI 1.01–1.20) for AVC by CT
Bouchareb, et al. [48]	2015	Lp(a) transports autotaxin to the AV which contributes to inflammation and calcification of the valve
Despres, et al. [49]	2019	In individuals without clinical AS, elevated Lp(a) associated with AV microcalcification by PET/CT
Zheng, et al. [50]	2019	Higher Lp(a) and OxPL levels associated with greater aortic valve calcification activity by PET/CT Lp(a) induces osteogenic differentiation of vascular cells, mediated by OxPL
**Lp(a) and AS**		
Glader, et al. [51]	2003	Lp(a) ≥ 48 mg/dL associated with increased risk for AS (OR 3.4, 95% CI 1.1–11.2)
Kamstrup, et al. [52]	2014	Lp(a) associated with AS in a graded fashion: >95th percentile of Lp(a) (>90 mg/dL): OR 2.9 (95% CI 1.8–4.9)
Arsenault, et al. [53]	2014	Top tertile of Lp(a) associated with increased risk for AS: HR 1.57, 95% CI 1.02–2.42
Langsted, et al. [54]	2015	Each 1-SD increase in Lp(a) associated with HR 1.23 (95% CI 1.06–1.41) for AS
**OxPL-apoB and AS**		
Kamstrup, et al. [55]	2017	Dose-dependent association between OxPL-apoB and CAVD For >95th percentile of levels, OR 3.4 (95% CI 2.1–5.5)
Que, et al. [56]	2018	Inactivation of OxPL reduces development of AV calcification and AV gradient in mice
**Lp(a), OxPL-apoB and AS Progression**		
Capoulade, et al. [57]	2015	Top tertile of Lp(a) (OR 2.6, 95% CI 1.4–5.0) and top tertile of OxPL-apoB (OR 2.4, 95% CI 1.2–4.6) associated with rapid AS progression
Capoulade, et al. [58]	2018	Lp(a) (OR 1.10, 95% CI 1.03–1.19 per 10 mg/dL increase) and OxPL-apoB (OR 1.06, 95% CI 1.01–1.12 per 1 nM increase) levels linearly associated with faster AS progression, especially in younger participants.
Zheng, et al. [50]	2019	Higher Lp(a) and OxPL levels associated with faster progression of AV calcium score by CT and hemodynamic progression by echocardiography
**Genetic Associations**		
Thanassoulis, et al. [6]	2013	rs10455872 associated with AVC in GWAS (OR per allele 2.05, *p* < 0.001) *LPA* genotype associated with incident AS (HR per allele 1.68, 95% CI 1.32–2.15) and AV replacement (HR 1.54, 95% CI 1.05–2.27)
Kamstrup, et al. [52]	2014	Genotypes corresponding with Lp(a) levels associated with increased risk of AS (HR 1.6, 95% CI 1.2–2.1 per 10-fold Lp(a) increase)
Arsenault, et al. [53]	2014	Carriers of rs10455872 SNP have increased risk of AS: Heterozygous: HR 1.78, 95% CI 1.11–2.87 Homozygous: HR 4.83, 95% CI 1.77–13.20
Langsted, et al. [54]	2015	Causal risk ratio for AS based on *LPA* SNPs (rs3798220, rs10455872): 1.38 (95% CI 1.23–1.55) Causal risk ratio for AS based on *LPA* KIV-2 genotype: 1.21 (95% CI 1.06–1.40)

AS = aortic stenosis, AV = aortic valve, AVC = aortic valve calcification, CT = computed tomography, GWAS = genome-wide association study, KIV = kringle 4, OxPL = oxidized phospholipids, OxPL-apoB = oxidized phospholipids on apolipoprotein B, SNP = single nucleotide polymorphism.

**Table 3 jcm-11-06040-t003:** Current and emerging therapies for Lp(a).

Current Therapies				
Drug	Target	Mechanism	Effect on Lp(a)	CVD Outcomes
Lipid Lowering Therapy				
Statins	HMG-CoA reductase	Inhibit cholesterol production	Do not lower Lp(a) levels, may increase Lp(a) [59]	Reduced ASCVD risk, but Lp(a) associated risk persists in statin treated individuals [28]
Ezetimibe	Nieman Pick C1-like1 protein	Reduces absorption of cholesterol in the small intestine	Limited data (possible 3–29% decrease in Lp(a)) [61,62]	No known effect on Lp(a)-associated risk
Niacin	Multifactorial	Downregulates *LPA* gene promotor and reduces apoB and triglycerides, increases HDL [63]	AIM-HIGH: 21% reduction in Lp(a), low absolute reduction [64] HPS2-THRIVE: low absolute reduction [65]	AIM-HIGH trial: no effect on CVD events [64]. HPS2-THRIVE: no overall effect of niacin on major vascular events [65]
Mipomersen	apoB	Anti-sense inhibitor of apoB synthesis	Reduces Lp(a) by median 26% [66]	Unclear effect on CV outcomes. Risk of liver toxicity
Lomitapide	Microsomal triglyceride transfer protein (MTP)	Inhibition of MTP inhibits transfer of lipids onto apoB	Reduces Lp(a) by 17% [61]	Unclear effect on CV outcomes. Risk of liver toxicity
PCSK9i (alirocumab, evolocumab, inclisiran)	PCSK9	Inhibit degradation of LDL-receptor	Reduce Lp(a) by 19–27% [67,68,69]	Limited data, however, reduction in Lp(a) associated with a reduction in CVD events (15% per 25 nmol/L in FOURIER, 0.6% per 1 mg/dL in ODYSSEY OUTCOMES) [67,68], but may not address inflammatory risk associated with OxPL [70]
Lipoprotein apheresis	apoB-containing lipoproteins	Removal of apoB-containing lipoproteins from plasma	Immediate reduction in Lp(a) levels up to 75%, with 30–35% time-averaged reduction when performed every 1–2 weeks [71]	Reduction in Lp(a) and LDL-C translates into significant reduction in MACE events in observational studies [72,73] MultiSELECt is an ongoing multicenter prospective study [74]
**Anti-platelet therapy**				
Aspirin	COX (cyclooxygenase) [75]	Reduces platelet aggregation through irreversible inhibition of thromboxane A_2_	--	In White women carriers of *LPA* rs3798220 SNP, aspirin associated with significant reduction in CVD risk (HR 0.44, 95% CI 0.20–0.94) in the Women’s Health Study [75]. Similar results in the ASPREE trial with the same SNP or high genetic risk score [76].
Dual anti-platelet therapy (DAPT)	Multifactorial	Multifactorial	--	In CAD patients with Lp(a) >30 mg/dL who underwent PCI, DAPT >1 year resulted in a significant reduction in CVD events (HR 0.54, 95% CI 0.31–0.92) compared with DAPT ≤1 year [77]
**Other**				
Hormone replacement therapy (estrogen)	--	Possibly through increased Lp(a) uptake by LDL receptor or decreased Lp(a) production [78]	Reduction in Lp(a) of 7.9 nmol/L [79]	No impact on CHD events
L-carnitine	--	Possibly related to fatty acid oxidation	Reduction in Lp(a) of 8.8 mg/dL [80]	Unclear effect on CV outcomes L-carnitine associated with increased CVD risk [81]
**Emerging Therapies**				
**Drug**	**Target**	**Mechanism**	**Effect on Lp(a)**	**Current stage in development**
Pelacarsen	apo(a) mRNA	Antisense oligonucleotide (ASO), binds apo(a) mRNA, targets for degradation	Phase I: Dose-dependent, up to −77.8% [82,83]. Ligand-conjugated form: dose-dependent, up to −92% [83] Phase II: Ligand-conjugated form: dose-dependent, up to −80% [84]	Phase III/cardiovascular outcomes trial underway (80 mg monthly subcutaneous injection vs. placebo) (NCT04023552).
Olpasiran	apo(a) mRNA	Small interfering RNA (siRNA), binds apo(a) mRNA, targets for degradation	Phase I: Maximum mean percent change in Lp(a) from baseline: −71% to −97% [85]	Phase II underway (NCT04270760)
SLN360	apo(a) mRNA	siRNA, binds apo(a) mRNA and targets for degradation	Phase I: Maximal median percent reduction in Lp(a), dose-dependent, up to −98% [86]	Phase II planned for 2022 [87]

apoB = apolipoprotein B, ASCVD = atherosclerotic cardiovascular disease, CAD = coronary artery disease, CHD = coronary heart disease, CVD = cardiovascular disease, HDL = high-density lipoprotein, LDL = low-density lipoprotein, mRNA = messenger ribonucleic acid, PCSK9 = proprotein convertase subtilisin/kexin type 9.

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
