# Peer review of "Lipoprotein(a): Evidence for Role as a Causal Risk Factor in Cardiovascular Disease and Emerging Therapies"

_jcm, 2022, doi:10.3390/jcm11206040_

Round 1
Reviewer 1 Report
1. Authors can include a table of lipoproteins involved in various cardiovascular diseases. Please update.
2. What are the causes behind lipoprotein deposition? Please include the details about this.
3. Is the lipoprotein associated with other diseases too? Please discuss and update.
4. Please discuss more lipoprotein and cardiovascular inflammation. What are the other immunologic aspects of lipoprotein deposition?
5. Authors need to discuss the mechanisms of lipoproteins in disease pathogenesis and development.
6. There are a lot of opportunities to include illustrations to make the review more interactive. Please update.
7. What are the early diagnostic methodologies for the detection of lipoproteins levels in the body? Please discuss this.
Reviewer 2 Report
Dear Author(s),
Thank you for this interesting, comprehensive and well-written manuscript.
Epidemiological findings consistently indicate direct and dose-dependent risk association of Lp(a) with atherosclerotic cardiovascular disease and calcific aortic valvular disease. Thus, this is a relatively novel and highly important topic for current and further clinical practice.
To the best of my knowledge, all relevant epidemiologic and genetic studies regarding Lp(a) and risk of ASCVD and aortic stenosis/calcific aortic valvular disease, are included in the manuscript and their key findings are accurately and clearly presented and highlighted throughout the text and the tables.
What is more, relevant data on current and emerging pharmacological therapeutic approaches is nicely summarized and well-presented. While expecting, phase III trials on the efficacy of some emerging agents, understanding and disseminating the current knowledge from earlier clinical trial phases is highly important to set the basis for the time (that is coming very soon) when therapeutic armamentarium to combat Lp(a) and consequently ASCVD and even maybe calcific aortic valvular disease is going to be available on the market (and when we will be able to start collecting even RWE and (cost) effectiveness data).
To deduce, since, the article is well-written I only have two potential suggestions/comments.
(I) Maybe you can also consider implementing some data regarding the level of (modest) effectiveness of non-pharmacological interventions (e.g. weight loss, diet type, exercise type and intensity) in terms of reducing Lp(a)?
(II) Also, maybe you can consider providing some comments and data regarding l-carnitine supplementation effectiveness as well?
Best regards, Reviewer
Round 2
Reviewer 1 Report
The authors responded just as a summary of the reviewer's comments rather than a point-to-point response.
The authors seem to be overlooked and are not willing to update the MS as per the reviewer's comments, however, this is a review article and there is always a chance to improve.